# ForesightKV: Optimizing KV Cache Eviction for Reasoning Models by Learning Long-Term Contribution

Zican Dong[1]  Peiyu Liu[2]  Junyi Li[3]  Zhipeng Chen[1]  Han Peng[1]  Shuo Wang[4]  Wayne Xin Zhao[1][†]

## Abstract

Recently, large language models (LLMs) have shown remarkable reasoning abilities by producing long reasoning traces. However, as the sequence length grows, the key-value (KV) cache expands linearly, incurring significant memory and computation costs. Existing KV cache eviction methods mitigate this issue by discarding less important KV pairs, but often fail to capture complex KV dependencies, resulting in performance degradation. To better balance efficiency and performance, we introduce ForesightKV, a training-based KV cache eviction framework that learns to predict which KV pairs to evict during long-text generations. We first design the Golden Eviction algorithm, which identifies the optimal eviction KV pairs at each step using future attention scores. These traces and the scores at each step are then distilled via supervised training with a Pairwise Ranking Loss. Furthermore, we formulate cache eviction as a Markov Decision Process and apply the GRPO algorithm to mitigate the significant language modeling loss increase on low-entropy tokens. Experiments on AIME2024 and AIME2025 benchmarks of three reasoning models demonstrate that ForesightKV consistently outperforms prior methods under only half the cache budget, while benefiting synergistically from both supervised and reinforcement learning approaches. Code is available at https://github.com/RUCAIBox/ForesightKV.

† Corresponding author. [1] Gaoling School of Artificial Intelligence, Renmin University of China [2]University of International Business and Economics [3]Department of Data Science, City University of Hong Kong [4]Tsinghua University. Correspondence to: Wayne Xin Zhao <batmanfly@gmail.com>.

*Proceedings of the $43^{rd}$ International Conference on Machine Learning*, Seoul, South Korea. PMLR 306, 2026. Copyright 2026 by the author(s).

## 1. Introduction

Recently, large language models (LLMs), especially the reasoning models, have demonstrated exceptional long-context generation capacities (Comanici et al., 2025; DeepSeek-AI et al., 2025; Zhao et al., 2023; Chen et al., 2025). With the extension of context windows and advancements in large-scale reinforcement learning, these models can generate complex and long reasoning paths for reasoning tasks, enabling them to solve problems through step-by-step reasoning (DeepSeek-AI et al., 2025; Shao et al., 2024; Peng et al., 2024). However, as the model's output length increases, the size of the model's internal key-value (KV) cache grows linearly, leading to decreased generation speed and increased memory overhead. Taking Qwen3-4B (Yang et al., 2025a) as an example, at a sequence length of 32K, the KV cache storage for a single instance consumes 4.5 GB with the precision of BFloat16, severely limiting the number of concurrent batches. Furthermore, since decoding is inherently memory-bound, loading an excessively large KV cache incurs significant latency (Yuan et al., 2024).

To address the overhead of the KV cache in long texts, numerous KV cache compression methods have been proposed, which reduce costs by either permanently reducing the number of KV pairs or reducing the overhead of storing a single KV pair (Li et al., 2024; Zhang et al., 2023; Liu et al., 2024b; Wang et al., 2024; Chang et al., 2024). Among them, the majority are designed for long-input tasks, with only a few studies being applicable to long-text generation. During the generation process, these methods typically employ elaborately designed rules (*e.g.,* attention scores, features of the KV pairs, and their positions) to estimate the importance of KV pairs and permanently evict unimportant ones at each eviction step (Cai et al., 2025; Zhang et al., 2023; Xiao et al., 2024; Wu et al., 2024). Thus, in spite of the linearly growing KV cache of a standard Transformer, under the fixed-budget eviction setting the retained cache is always bounded by the preset budget throughout decoding. However, these training-free methods are often insufficient to take all the complex patterns across different attention heads into account, typically leading to suboptimal performance. In contrast, other works employ training-based methods to perform a one-time evaluation of KV pair importance for

eviction (Lancucki et al., 2025), which fails to capture the dynamic nature of their importance at different stages of the sequence.

To achieve a better tradeoff between generation quality and efficiency in the KV cache eviction process, we first investigate the behavior of Qwen3-4B (Yang et al., 2025a) on questions and reasoning traces generated by itself. We observe that KV pairs across different attention heads often exhibit different patterns, which can be categorized into three main types, *i.e.,* global, position-dependent, and semantic-dependent (as shown in Figure 1). Among them, due to the inherent properties of reasoning data, semantic-dependent patterns exhibit greater complexity compared to the other two types, manifesting features such as block-wise attention structures and dynamic variations. This makes it difficult to capture them with rule-based algorithms or methods that determine importance in a single pass (Zhang et al., 2023; Lancucki et al., 2025). In addition, we observe that the losses of low-entropy tokens sharply change due to the absence of critical KV pairs, resulting in factual errors that may distort subsequent reasoning. These observations highlight the necessity of eviction methods that can adaptively track semantic dependencies and mitigate the significant loss increase of low-entropy tokens during generation.

In this work, we propose ForesightKV, a method that learns to predict the long-term contribution of each KV pair and adaptively evicts less important ones during generation. The key idea is to train a lightweight scoring model that estimates the dynamic importance of each KV pair and guides eviction decisions. To achieve this, ForesightKV employs a two-stage training paradigm: supervised learning with constructed future importance labels, followed by reinforcement learning to refine eviction policies. We first introduce the *Golden Eviction* algorithm to construct supervision labels. Specifically, we partition the attention matrix of a sequence into fixed-length blocks along the query dimension and aggregate attention scores within each block and attention heads within the same group to compute block scores. For each eviction step, we employ the maximum block scores of future steps as future scores and discard pairs with the lowest future scores, thus minimizing the impact of eviction. The scoring model is then trained with these labels using a pairwise ranking loss to capture relative eviction preferences. Subsequently, we model the KV cache eviction as a Markov Decision Process (MDP) and optimize the scoring models with reinforcement learning. On a full reasoning sequence, the scoring model selects eviction actions at each step, which results in different eviction traces. We formulate the sequence reward based on the MSE loss of language modeling losses before and after KV cache eviction for tokens that are initially predicted with low entropy and with large loss increases. The scoring models are then trained using the GRPO algorithm to maximize this

cumulative reward. Notably, our framework trains solely the lightweight scoring models with no backward operations or parameter updates on the LLMs.

To evaluate the effectiveness of our method, we compare ForesightKV with other KV cache eviction methods on three reasoning LLMs using two math benchmarks, *i.e.,* AIME2024 and AIME2025. Experiments show that our method can achieve better results than the baselines with even only half the KV cache. Specifically, ForesightKV preserves 92% and 99% of the original model's performance under 2K and 4K budget constraints, respectively. Furthermore, our approach yields substantial throughput improvements while presenting superior generalization capabilities on similar long-form generation tasks.

## 2. Empirical Study

### 2.1. Attention Patterns During Long-Text Reasoning

To investigate the properties of the KV cache during long-context reasoning, we employ Qwen3-4B (Yang et al., 2025a) to generate responses to questions from the STILL dataset (Min et al., 2024). We then select a representative response to calculate attention scores and visualize the attention maps of three attention heads, as illustrated in Figure 1. Consistent with the findings of MInference 1.0 (Jiang et al., 2024), we observe distinct attention patterns across different attention heads. Based on the dynamic characteristics of these attention patterns, we categorize the KV pairs into three types, namely global, position-dependent, and semantic-dependent.

• *Global*: Global KV pairs manifest as vertical lines in the attention map. This visual pattern indicates these pairs receive a high attention weight from most queries.

• *Position-dependent*: In Transformer decoders, the attention mechanism often demonstrates locality, with most of the attention allocated to tokens near the query (Xiao et al., 2024). As the decoding length increases, attention to earlier KV pairs decreases, reducing their overall importance.

• *Semantic-dependent*: In contrast to the other two types, semantic-dependent KV pairs exhibit more complex properties. As shown in Figure 1 (C), the attention map often displays a block-wise pattern due to inherent structural features in long-text reasoning data. In this case, attention is primarily concentrated on KV pairs within the current block and in previously semantically related blocks. As decoding moves into subsequent blocks, the high-attention regions may shift accordingly. Moreover, after a specific query, KV pairs that previously received high attention may become permanently irrelevant, as illustrated in Figure 1 (D).

Given the complexity of these attention mechanisms, previous KV cache eviction algorithms often fail to capture

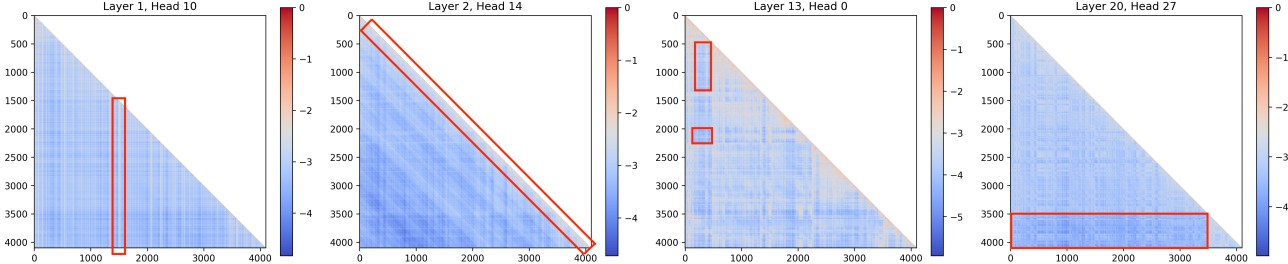

*Figure 1.* KV patterns in Qwen3-4B, including three patterns: (A) Global; (B) Position-dependent; (C) and (D) Semantic-dependent.

such intricate patterns. For instance, at each eviction step, SnapKV (Li et al., 2024) uses the last few tokens as an observation window and utilizes their attention scores to evaluate the importance of KV pairs. However, many semantic-dependent KV pairs will often exhibit different importance as the semantics change. Therefore, SnapKV often discards tokens that have little semantic association with the tokens of that window but are subsequently important, causing severe performance loss. Furthermore, these patterns are not mutually exclusive and can co-occur within a single attention head, posing an additional challenge for KV cache eviction design. In Appendix C, we present the ability of different methods to capture attention characteristics.

### 2.2. Large Shifts in Loss for Low-Entropy Tokens

In reasoning tasks, the entropy of LLMs plays a crucial role. High-entropy tokens (tokens with top-20% entropy) often mark decision points between different reasoning paths, while low-entropy tokens (other tokens) make up the more detailed and deterministic parts of the reasoning process (Wang et al., 2025). To investigate how these distinct token types behave, we employ Qwen3-4B (Yang et al., 2025a) to compute the loss on given sequences and compare the changes in loss for high-entropy and low-entropy tokens with R-KV (Cai et al., 2025) with the budgets of 1024 tokens across math, coding, and summarization tasks. As shown in Table 1, low-entropy tokens are influenced largely by the KV cache eviction, which is reflected as a greater proportional increase in the loss. We infer that this may be due to the eviction of contextual information, which is crucial for maintaining the low entropy of these tokens (Qiu et al., 2025). Furthermore, we analyze prediction errors on low-entropy tokens and find they often involve numbers, symbols, and entities that have appeared in the previous context (see Appendix D), whose factual mistakes can accumulate over long sequences and distort subsequent reasoning. Thus, it is very important to consider these low-entropy tokens for KV cache eviction.

*Table 1.* Loss ratio of tokens with different entropies after eviction.

| Entropy | Math | Code | Summarization |
|---|---|---|---|
| Low-Entropy | +147% | +75% | +187% |
| High-Entropy | +52% | +1% | +142% |

## 3. ForesightKV

To effectively manage the KV cache in long-context generation, we introduce **ForesightKV**, a training-based eviction method that balances memory efficiency with generation quality. The key idea is to learn a scoring model to predict the long-term contribution of KV pairs and guide eviction decisions under a fixed memory budget. Specifically, our approach begins by formally defining the compression process during long-context generation with scoring models in our framework (Section 3.1). We then propose a two-stage training pipeline consisting of supervised training (Section 3.2) and reinforcement learning (Section 3.3) to optimize the scoring models' long-term prediction capacities. A flowchart of our method is presented in Figure 2.

### 3.1. Dynamic KV Cache Eviction During Long-Context Generation

In general, we define a dynamic compression process in our framework, parameterized by two hyperparameters: a cache budget $B$, which specifies the target number of KV pairs to retain, and an eviction length $L$, which determines the frequency of the eviction process. The eviction is triggered periodically: after every $L$ new tokens are generated, the cache size reaches $B + L$. At this point, we keep the most recent $L$ KV pairs and employ an eviction algorithm that prunes the other cache back to the budget size $B - L$ by retaining only the most important KV pairs. Specifically, at each eviction step, we assign a scoring model $\pi_\theta$ to evaluate the importance of each KV pair. To balance efficiency and performance, we adopt an MLP scorer $\pi_\theta$ to predict the importance score $\phi_n$ of the $n$-th KV pair. Motivated by the complex and diverse KV cache patterns observed in Section 2.1, we construct the input feature $\mathbf{x}_n$ by concatenating the key $\mathbf{k}_n$, the value $\mathbf{v}_n$, and their associated

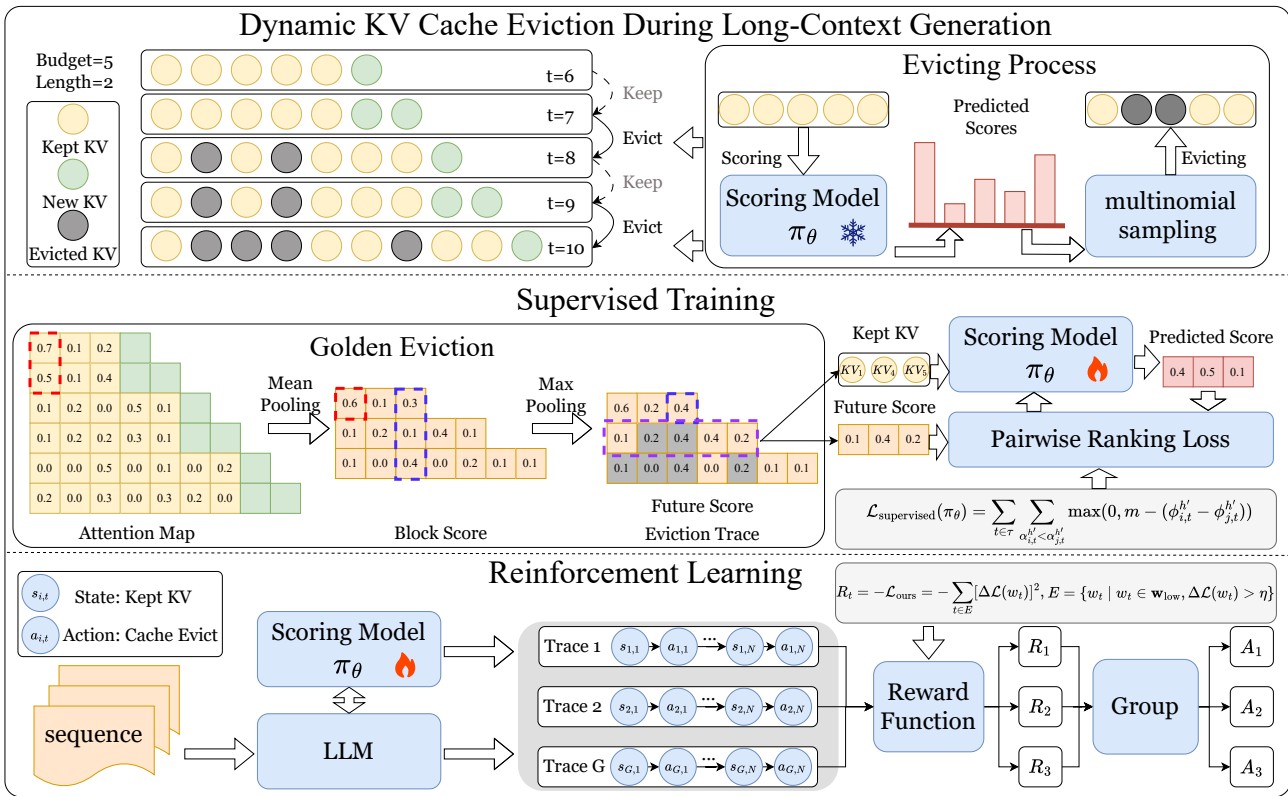

*Figure 2.* Overview of ForesightKV. ForesightKV uses a scoring model to guide dynamic KV cache eviction during long-context generations, which is trained through supervised learning and then reinforcement learning. The dashed box indicates pooling.

attention features $\mathbf{a}_n$, fixed-length vectors transformed from the attention scores (see Appendix B).

$$\mathbf{x}_n = \text{Concat}(\mathbf{k}_n, \mathbf{v}_n, \mathbf{a}_n), \quad \phi_n = \pi_\theta(\mathbf{x}_n). \quad (1)$$

Based on the importance scores $\Phi = \{\phi_n\}$, we parameterize the eviction action with a Top-K multinomial sampling strategy. Specifically, we first construct a high-confidence candidate set by selecting the $2L$ KV pairs with the lowest predicted importance scores, and then sample $L$ pairs from this set according to the normalized negative scores:

$$\mathcal{D}_t = \text{Multinomial}_L \left( \text{Softmax} \left( \text{Top}_{2L}(-\Phi) \right) \right), \quad s'_t = s_t \backslash \mathcal{D}_t. \quad (2)$$

This design is well aligned with the sequential and irreversible nature of KV cache eviction. Pure top-$K$ eviction behaves like greedy decoding: it is stable but can be sensitive to local ranking errors made by the scoring model. In contrast, pure multinomial sampling provides stronger exploration but is often too noisy when an erroneous eviction cannot be recovered in later steps. Top-K multinomial sampling first restricts the action space to low-importance candidates and then performs controlled sampling within this set, achieving a better balance between stability and exploration. This action parameterization improves inference-time robustness and also facilitates reinforcement learning

by preserving useful stochasticity. Table 5 demonstrates the superiority of this method over other pure alternatives.

We provide the details of the scoring model and sampling method in Appendix B. In the following, we present the two-stage training framework for the scoring models. In the first stage, we employ the "Golden Eviction" algorithm that leverages future information to generate optimal eviction traces, which serve as supervision for training the scoring model. In the second stage, we refine the model via reinforcement learning by formulating eviction as a Markov Decision Process. Throughout training, the LLM parameters remain frozen, and only the lightweight scoring models are updated.

### 3.2. Supervised Training

In our two-stage framework, we begin with a supervised training phase to equip the scoring model with the ability to identify critical KV pairs. This stage is motivated by our analysis (Section 2), which shows that the relevance of a KV pair evolves dynamically during generation. To predict the future importance, we introduce the *Golden Eviction* algorithm, which constructs target labels and eviction traces. The scoring models are then trained on these labels using a Pairwise Ranking Loss.

**Golden Eviction.** As proved in Appendix A, discarding KV pairs with the lowest attention scores has the minimal impact on the upper bound of the model's output. Inspired by this observation, we propose the "Golden Eviction" algorithm. Specifically, for a given full reasoning trace consisting of the prompt and generated response, $\mathbf{w} = \{w_1, \ldots, w_T\}$, we compute the corresponding attention score matrix $\mathbf{A}^h \in \mathbb{R}^{T \times T}$ on the original model, where $h$ denotes head index on given layer. Subsequently, we partition the attention matrix along the query dimension with a stride of $L$, starting from the first eviction position $B + L$, where the final block is padded to match the maximum length. We employ pooling across the query dimension and attention group to obtain the block scores $\tilde{\mathbf{a}}_t^h$ of all blocks for each kv pair:

$$\mathbf{a}_t^h = \text{Pool}(\mathbf{A}_{B+tL:B+(t+1)L-1,:}^h) \in \mathbb{R}^T, \qquad (3)$$

$$\tilde{\mathbf{a}}_t^{h'} = \text{Pool}(\{\mathbf{a}_t^{g(h'-1)+1}, \ldots, \mathbf{a}_t^{gh'}\}) \in \mathbb{R}^T, \qquad (4)$$

where $t \in \{1, \ldots, M\}$ is the number of eviction steps and $M = \lceil (T - B)/L \rceil - 1$. For simplicity, we employ average pooling for the two pooling computations. For each eviction step $t$, our objective is to minimize the impact on future attention computations. In other words, we want the evicted KV pairs to have low attention scores in all future blocks. Thus, we first compute the maximum block score over all future blocks as the future score $\alpha_{i,t}^{h'} \in \mathbb{R}$ for the $i$-th KV pair, and only keep the KV pairs with the largest future scores as the retained cache $s_t^{h'}$.

$$\alpha_{i,t}^{h'} = \max_{t \leq j \leq M} (\tilde{\mathbf{a}}_{i,j}^{h'}), \quad s_t^{h'} = \text{Top}_{B-L}(\alpha_{i,t}^{h'}). \qquad (5)$$

**Training.** Subsequently, based on the KV cache eviction traces obtained from the Golden Eviction algorithm, we employ the scoring model to compute a score for each KV pair at every eviction step. Furthermore, we frame the eviction as a ranking task and use Pairwise Ranking Loss to train the scoring models. Our objective is to ensure that the model's predicted scores for any two KV pairs are ranked in the same order of their future scores. For any pair of indices $i$ and $j$, if $\alpha_{i,t}^{h'} > \alpha_{j,t}^{h'}$, we require that $\phi_{i,t}^{h'} > \phi_{j,t}^{h'}$. Formally, we define the training loss function as follows ($m$ is a hyper-parameter for the loss):

$$\mathcal{L}_{\text{supervised}}(\pi_\theta) = \sum_{t \in \tau} \sum_{\alpha_{i,t}^{h'} < \alpha_{j,t}^{h'}} \max(0, m - (\phi_{i,t}^{h'} - \phi_{j,t}^{h'})). \qquad (6)$$

**Effectiveness of Golden Eviction.** To demonstrate the effectiveness of the Golden Eviction algorithm, we compare it against three leading KV cache eviction algorithms: R-KV (Cai et al., 2025), SnapKV (Li et al., 2024), and H2O (Zhang et al., 2023). Using Qwen3-4B (Yang et al.,

2025a) models, we measure the percentage increase in model loss on sampled inference data. We evaluate these methods with the KV cache budgets of 1024 and 2048, and employ the KV cache eviction every 128 or 256 tokens. As detailed in Table 2, Golden Eviction consistently yields significantly lower loss than the competing approaches. These results confirm the effectiveness of Golden Eviction in preserving model performance under strict memory constraints.

*Table 2.* Comparison of model loss ratios for different methods relative to the original model under various KV cache budgets. Here, (1024,128), (2048,128), (1024,256), and (2048,256) denote KV cache budgets of 1024 and 2048, with KV cache evicts every 128 or 256 tokens.

| Method | (1024,256) | (2048,256) | (1024,128) | (2048,128) |
|--------|-----------|-----------|-----------|-----------|
| Golden | **1.0711** | **1.0166** | **1.0715** | **1.0185** |
| R-KV | 1.4101 | 1.1606 | 1.4750 | 1.1814 |
| SnapKV | 1.4091 | 1.1281 | 1.4214 | 1.1343 |
| H2O | 1.2730 | 1.0948 | 1.4106 | 1.1578 |

### 3.3. Reinforcement Learning

Despite the effectiveness of supervised training with Golden Eviction, it fails to account for the distribution shift in internal states caused by the eviction during inference. Thus, the ideal eviction trace is different from the Golden Eviction algorithm. To bridge this gap between the supervised training and the dynamic inference process, we further refine the model using reinforcement learning.

For KV cache eviction in long-context decoding, the retained KV cache influences both the current generation of the model and subsequent cache eviction decisions. Therefore, we define KV cache eviction during inference as a Markov Decision Process (MDP) and employ reinforcement learning algorithms to optimize the eviction process. Given a full reasoning trace $\mathbf{w} = \{w_1, \ldots, w_T\}$, and the frozen LLM $\Theta$, our goal is to maximize the rewards by optimizing the set of scoring models $\{\pi_{\theta_{h,l}}\}$. In the following, we first define the components of the reinforcement learning framework:

● State ($s_t$): The current remaining KV cache at the step $t$.

● Action ($a_t$): Given a state $s_t$, the action $a_t$ is to select a subset of indices of the KV pairs $\{1, \ldots, B + L\}$ to retain for the current generation step by a predefined cache budget.

● Policy ($\pi_\theta$): For each attention group, we define the scoring model as the policy model, which assigns a score to guide whether the KV pair should be retained or discarded.

● Reward ($R_t$): To evaluate the quality of an eviction action, we define a sequence-level reward. Inspired by the observations in Section 2.2, we find that increases in the loss of low-entropy tokens have a substantial impact on reasoning quality. Based on this insight, we construct a

subset of tokens $E$ according to two criteria: (1) the token's original entropy falls within the bottom $80\%$ of the sequence $\mathbf{w}_{\text{low}}$, and (2) its loss increase from the original model ($\mathcal{L}_{\text{ori}}$) to the evicted model ($\mathcal{L}_{\text{evict}}$) exceeds a threshold $\eta$, $\Delta\mathcal{L}(w_t) = \mathcal{L}_{\text{ori}}(w_t) - \mathcal{L}_{\text{evict}}(w_t)$. The subset $E$ is formulated as follows:

$$E = \{w_t \mid w_t \in \mathbf{w}_{\text{low}}, \Delta\mathcal{L}(w_t) > \eta\}. \quad (7)$$

The reward is then defined as the average square of the loss increases $\Delta\mathcal{L}(w_t)$ across all tokens in $E$, formulated as:

$$R_t = -\mathcal{L}_{\text{ours}} = -\sum_{t \in E} [\Delta\mathcal{L}(w_t)]^2. \quad (8)$$

We use the GRPO algorithm (Shao et al., 2024) to train the scoring models. We initialize our policy model $\pi_\theta$ with the model obtained from supervised training, which also serves as the reference model $\pi_{\text{ref}}$. At each training step, for a given sequence $\mathbf{w}$, we sample $G$ KV cache eviction traces for all layers and heads using the Top-K multinomial sampling method with the old policy $\pi_{\theta_{\text{old}}}$, i.e., $\{o_1, \ldots, o_G\}$. Influenced by the different kept KV caches across traces, the same token will exhibit different hidden states, resulting in varying rewards throughout the entire sequence. Subsequently, we employ group relative normalization to compute the estimated advantage scores for these traces:

$$\hat{A}_t = (R_t - \text{Mean}(R_t))/\text{Std}(R_t). \quad (9)$$

Furthermore, we broadcast these advantage scores to every eviction process for all scoring models and optimize these models together using the following training objective.

$$\begin{aligned} \mathcal{J}(\theta) =& \mathbb{E}_{o \sim \pi_{\theta_{\text{old}}}} \Bigg[ \sum_{t=1}^{|o|} \min\left( r_t(\theta)\hat{A}_t, \right. \\ & \left. \text{clip}(r_t(\theta), 1 - \epsilon, 1 + \epsilon)\hat{A}_t \right) \\ & - \beta \cdot \text{KL}\left[\pi_\theta(a_t|s_t) \| \pi_{\text{ref}}(a_t|s_t)\right] \Bigg], \quad (10) \end{aligned}$$

where $r_t(\theta) = \frac{\pi_\theta(a_t|s_t)}{\pi_{\theta_{\text{old}}}(a_t|s_t)}$ is the importance sampling ratio, $\epsilon \in \mathbb{R}$ is the clipping threshold, and $\beta$ controls KL regularization.

## 4. Experiments

### 4.1. Experiments Setup

**Evaluated Models and Benchmarks.** To evaluate the effectiveness of our method on the long-text reasoning task, we choose DeepSeek-R1-Distill-Qwen-7B (DeepSeek-AI et al., 2025), Qwen3-4B and Qwen3-1.7B (Yang et al.,

2025a) as evaluated models and evaluate them on two complex math benchmarks, i.e., AIME2024 and AIME2025. Following the settings of previous work (DeepSeek-AI et al., 2025; Yang et al., 2025a), we set the temperature of $0.6$, top-$k$ of $20$, and top-$p$ of $0.95$. To ensure statistical reliability, we report the average scores of pass@1, where each benchmark is evaluated independently 32 times. For each method, we evaluate the performance under different cache budgets $B$, e.g., 1024, 2048, and 4096, with the eviction length $L$ of 256. For both models, we set the immediate size of scoring models to 16. We also select the top 512 KV pairs first and sample 256 pairs from them. We introduce the details of training setups in Appendix E.1.

**Baselines.** We compare our method with three KV cache eviction methods, i.e., SnapKV (Li et al., 2024), H2O (Zhang et al., 2023), and R-KV (Cai et al., 2025). Following the settings in R-KV, we compress the KV cache every $L$ steps for all these methods and apply SnapKV during the long-decoding phase, where it performs dynamic compression at each step by leveraging the attention within a fixed-size window. The details of these methods are provided in Appendix E.2.

### 4.2. Main Results

Figure 3 presents the performance comparison of ForesightKV and other KV cache eviction methods. Firstly, our method achieves better performance under the same KV cache budgets compared to other eviction methods across various benchmarks. Compared to other KV cache eviction methods, ForesightKV achieves comparable or even better performance with only half the KV cache. For example, on the AIME2024 dataset, ForesightKV with the Qwen3-4B model and a budget of only 1K outperforms R-KV with a 2K budget (54.5 vs. 44.8). Additionally, on certain benchmarks, ForesightKV can achieve comparable performances with the original model with a budget of 4K KV pairs. This demonstrates that ForesightKV learns to better predict the long-term importance of KV pairs, thereby preserving the model's long-text reasoning capabilities more effectively.

Secondly, our method can benefit from both supervised training and reinforcement learning stages. The first stage of supervised training endows the scoring models with a powerful predictive capability for importance, significantly surpassing existing rule-based methods on benchmarks. Furthermore, the second stage of reinforcement learning enhances the scoring models by optimizing their capabilities through optimization of the overall LLMs. By optimizing the eviction policy to maximize the predictive accuracy of critical future tokens across all positions, the model's reasoning performance can be further improved within a fixed KV cache budget.

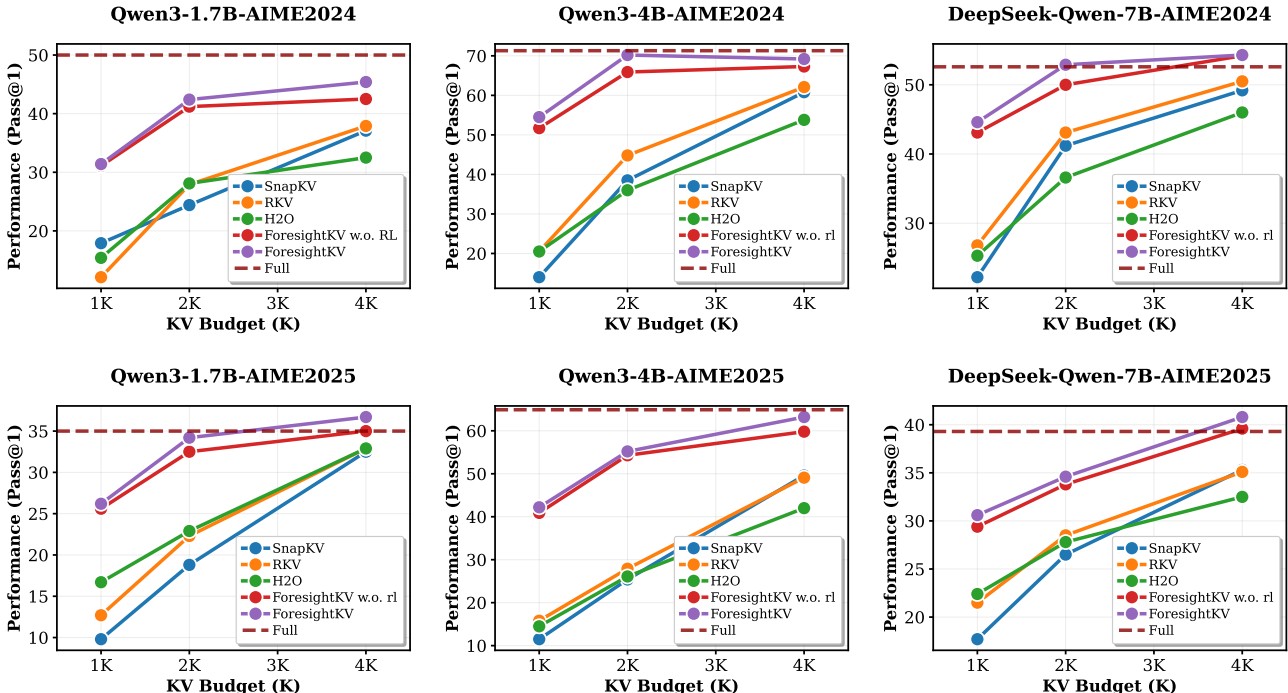

*Figure 3.* Comparison of ForesightKV with other KV cache eviction methods on reasoning tasks.

*Table 3.* Efficiency evaluation of ForesightKV. "#MCB" denotes the maximum concurrent batch size.

| Method | 8K | | | 16K | | | 32K | | |
|---|---|---|---|---|---|---|---|---|---|
| | #MCB | Throughput | Ratio | #MCB | Throughput | Ratio | #MCB | Throughput | Ratio |
| Full | 48 | 139.39 | 1.00× | 24 | 73.40 | 1.00× | 11 | 37.73 | 1.00× |
| ForesightKV-1K | 96 | 375.10 | 2.69× | 96 | 372.35 | 5.07× | 96 | 369.43 | 9.79× |
| ForesightKV-2K | 70 | 272.48 | 1.95× | 70 | 270.65 | 3.69× | 70 | 268.36 | 7.11× |
| ForesightKV-4K | 48 | 198.58 | 1.42× | 48 | 196.32 | 2.67× | 48 | 193.95 | 5.14× |
| ForesightKV-8K | - | - | - | 36 | 114.32 | 1.56× | 36 | 112.99 | 3.03× |

Finally, our method exhibits robust generalization across varying budgets. Although trained with restricted budgets ($B \leq 2K$) in both stages, it seamlessly generalizes to other settings, *e.g.,* a budget of 4K. This indicates that our approach captures the intrinsic importance of KV cache entries, rather than simply overfitting to specific budget constraints.

For experiments with more models and baselines, we present them in E.6 and Appendix E.8.

### 4.3. Further Analysis

**Efficiency of ForesightKV.** To evaluate the efficiency of ForesightKV on long-context generation tasks, we set the KV budget to 1K, 2K, and 4K tokens and compare it with the full KV cache. We measure the maximum concurrent batch size and throughput of Qwen3-4B on one A800 GPU with generation lengths of 8K, 16K, and 32K tokens. As shown in Table 3, applying ForesightKV can keep a fixed

batch size and throughput under different generation lengths. Additionally, it substantially increases the feasible batch size during generation and leads to significant throughput improvements. For instance, ForesightKV with a budget of 1024 tokens achieves up to a 9.79× throughput gain on long-context generation of 32K tokens. In Appendix E.9, we further demonstrate the efficiency of the scoring models and significant end-to-end speedup.

**Reward Design.** To optimize the KV cache eviction, we design and evaluate several distinct reward functions. (1) $\mathcal{L}_{\text{all}}$, $\mathcal{L}_{\text{low}}$, and $\mathcal{L}_{\text{high}}$: the language modeling losses for all, low-entropy, and high-entropy tokens, respectively; (2) $\mathcal{L}_{\text{low,large}}$: the loss on the subset of low-entropy tokens whose loss increases by more than 1.5 post-eviction; and (3) $\mathcal{L}_{\text{ours}}$: the MSE loss on this same subset of severely impacted tokens as $\mathcal{L}_{\text{low,large}}$. We use the reinforcement learning framework on Qwen3-4B with the budget size of $1K$ and present

the results in Table 4. Our findings reveal that naively minimizing the overall language modeling loss ($\mathcal{L}_{\text{all}}$) does not guarantee better performance. While targeting low-entropy tokens ($\mathcal{L}_{\text{low}}$) yields marginal gains, focusing on high-entropy ones ($\mathcal{L}_{\text{high}}$) causes significant degradation. The most effective strategy is to selectively optimize for low-entropy tokens that are most adversely affected by eviction. Using $\mathcal{L}_{\text{low,large}}$ and $\mathcal{L}_{\text{ours}}$ as reward signals reduces both the number of tokens with large loss spikes and the overall model perplexity. Notably, the MSE-based reward, $\mathcal{L}_{\text{ours}}$, is particularly effective at penalizing catastrophic loss increases, leading to the best final performance.

*Table 4.* Experiments of rewards with 1K budget.

| Reward | - | $-\mathcal{L}_{\text{all}}$ | $-\mathcal{L}_{\text{low}}$ | $-\mathcal{L}_{\text{high}}$ | $-\mathcal{L}_{\text{low,large}}$ | $-\mathcal{L}_{\text{ours}}$ |
|---|---|---|---|---|---|---|
| AIME24 | 51.7 | 50.6 | 53.5 | 49.6 | 53.8 | **54.5** |
| AIME25 | 40.9 | 40.0 | 40.4 | 35.4 | **42.3** | **42.3** |

**Design of Model and Sampling Algorithm.** We conduct ablation studies to analyze the effects of the scoring model's inputs and the sampling algorithms. Instead of concatenating attention features with KV representations, we evaluate the scoring model with only attention features. We also replace our sampling method (multinomial sampling from the top-K pairs) with either top-K or multinomial sampling. As shown in Table 5, evaluations on AIME2024 and AIME2025 with Qwen3-4B reveal that both modifications consistently degrade performance: top-K and multinomial sampling led to a significant performance drop, while only using attention features can hardly accurately predict the importance of the KV pairs.

*Table 5.* Ablation study of scoring model and sampling function with 1K budget. *Attn* and *KV* denote using attention features and KV representations. *Top-K* and *MN* denote the Top-K and multinomial sampling methods.

| Input | Sampling | AIME24 | AIME25 |
|---|---|---|---|
| Attn+KV | Top-K+MN | **51.7** | **40.9** |
| Attn | Top-K+MN | 37.5 | 22.9 |
| Attn+KV | MN | 16.5 | 13.8 |
| Attn+KV | Top-K | 46.0 | 37.7 |

**Generalization Capabilities on Reasoning Tasks.** To evaluate the generalization capabilities of ForesightKV, we conduct experiments on reasoning benchmarks in other domains, specifically GPQA (Rein et al., 2023) (science) and LiveCodeBench-V3 (Jain et al., 2025) (coding). The experimental results with Qwen3-4B are shown in Table 6. Similar to observations on math benchmarks, ForesightKV consistently outperforms all baselines on both datasets with only half the KV budgets, while the performance of ForesightKV approaches that of the original model under a limited budget.

*Table 6.* Generalization capacities evaluations of Qwen3-4B on GPQA and LiveCodeBench.

| Method | GPQA | | | LiveCodeBench | |
|---|---|---|---|---|---|
| | 1K | 2K | 4K | 1K | 2K |
| Full | | 54.6 | | | 63.4 |
| H2O | 25.2 | 29.7 | 43.1 | 28.3 | 42.5 |
| SnapKV | 16.9 | 34.8 | 49.1 | 27.0 | 48.1 |
| R-KV | 23.0 | 40.0 | 50.7 | 34.4 | 52.0 |
| ForesightKV(*w/o* RL) | 44.2 | 51.2 | 52.4 | **55.7** | **61.5** |
| ForesightKV | **45.2** | **51.3** | **53.7** | **55.7** | 61.1 |

**Generalization Capabilities on Long-Input Tasks.** We further evaluate whether ForesightKV generalizes to long-input tasks on LongBench (Bai et al., 2024). These experiments are conducted with Qwen3-4B under a 1K KV budget in the non-thinking mode. Since LongBench mainly consists of long-input and short-output tasks, we perform KV cache compression only once after prefilling, rather than repeatedly evicting KV pairs during decoding. As shown in Table 7, ForesightKV achieves the best overall average among all compression methods and remains close to the full model, especially on single-document QA, multi-document QA, summarization, and code completion. Given that the scoring models were trained exclusively on math tasks, these results demonstrate the superior generalization capabilities of ForesightKV beyond the training domains.

*Table 7.* Performance comparison of Qwen3-4B on LongBench task categories with a 1K KV budget in the non-thinking mode. KV cache compression is performed only once after prefilling.

| Task Category | Full Model | R-KV | SnapKV | H2O | ForesightKV |
|---|---|---|---|---|---|
| Single-document QA | 41.88 | 37.39 | 37.29 | 35.26 | **41.07** |
| Multi-document QA | 43.97 | 42.45 | 42.56 | 40.51 | **43.46** |
| Summarization | 27.48 | 24.08 | 24.97 | 21.13 | **27.39** |
| Few-shot Learning | 62.73 | 59.39 | **60.71** | 59.27 | 60.48 |
| Synthetic Tasks | 48.88 | **50.02** | 50.02 | 48.50 | 49.55 |
| Code Completion | 2.29 | 1.91 | 1.64 | 3.17 | **3.41** |
| Overall Average | 39.41 | 37.11 | 37.50 | 35.74 | **38.95** |

## 5. Related Work

**KV Cache Eviction.** To mitigate the memory and computational overhead of the KV cache, which scales linearly with context length, various techniques have been proposed to enhance model efficiency, including KV cache eviction (Zhang et al., 2023), merging (Wang et al., 2024), dynamic loading (Tang et al., 2024), low-rank decomposition (Chang et al., 2024), and quantization (Hooper et al., 2024; Liu et al., 2024b;a). Among these, KV cache eviction methods exploit the inherent sparsity of attention mechanisms to permanently discard less important KV pairs based on predefined rules. These methods typically rely on heuristic rules, *e.g.,* positions, attention scores, and representations of KV (Xiao et al., 2024; Zhang et al., 2023; Cai et al., 2025;

Li et al., 2024; Wu et al., 2024; Goel et al., 2025). However, such heuristic-based approaches are often suboptimal for long-generation tasks. Additionally, some work uses trainable methods to discard unimportant caches (Lancucki et al., 2025; Zeng et al., 2024; Huang et al., 2024). These methods make a one-time judgment on the importance of the KV cache and then permanently discard it, which is difficult to capture the changing importance as the sequence evolves. Our method is the first to jointly use supervised and reinforcement learning to optimize the eviction process. It dynamically captures the importance of KV pairs to decide on their eviction, leading to improved performance.

**Efficient Long-Context Generation.** With the extension of context windows and the enhancement of long-context abilities of LLMs, long-text generation tasks, particularly for long-reasoning tasks, have emerged as a focal point of research (DeepSeek-AI et al., 2025; Dong et al., 2025). After reinforcement learning, the model accurately solves problems by generating a lengthy reasoning process (Yang et al., 2025a). To further improve efficiency, some studies have reduced the model's generation length by designing reinforcement learning algorithms (Cheng et al., 2025) and employing model merging (Wu et al., 2025). Other studies focus on reducing the computational cost associated with generating each token via speculative decoding (Yang et al., 2025b) or KV cache compression (Cai et al., 2025), without changing the generation lengths. Our work falls into the latter category, improving efficiency by reducing overhead through KV cache eviction.

## 6. Conclusion

In this paper, we introduce ForesightKV, a novel two-stage training method for KV cache eviction tailored for long-context reasoning tasks. In the first stage, we design a Golden Eviction algorithm to obtain ideal eviction data, which is used to train scoring models with a Pairwise Ranking Loss. In the second stage, we model the eviction task as a Markov Decision Process and apply reinforcement learning to mitigate the significant performance loss on low-entropy tokens that can occur after eviction. Experiments on two reasoning benchmarks demonstrate that ForesightKV outperforms leading baselines with only half the cache budget. Ablation studies confirm that both the supervised and reinforcement learning stages are crucial to its success. We believe this reinforcement learning perspective opens up a promising new research direction for adaptive KV cache management in long-context reasoning.

## Impact Statement

This paper presents work whose goal is to advance the field of Machine Learning. There are many potential societal consequences of our work, none of which we feel must be specifically highlighted here.

## Acknowledgments

This work was partially supported by the National Natural Science Foundation of China under Grant No. 92470205 and 62506077, and Beijing Major Science and Technology Project under Contract No. Z251100008425002. Xin Zhao is the corresponding author.

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

# A. Theoretical Analysis of KV Cache Eviction

In this section, we analyze the approximation error induced by KV cache eviction and clarify its connection to Golden Eviction. We first study the error at the level of a single future query, showing that the perturbation is controlled by the evicted attention mass. We then connect this single-query result to the future-block surrogate used by Golden Eviction and explain why this surrogate is aligned with generation quality preservation.

**Problem Setup.** Let the original full attention output be $\mathbf{o}_t = \sum_{i=1}^N a_{t,i} \mathbf{v}_i$, where $a_{t,i} = \frac{\exp(s_{t,i})}{\sum_{k=1}^N \exp(s_{t,k})}$ are the original normalized attention scores. When evicting a set of indices $\mathcal{E}$, the remaining scores are renormalized. Let $\mathcal{S}$ be the retained set. The new output $\hat{\mathbf{o}}_t$ is:

$$\hat{\mathbf{o}}_t = \sum_{i \in \mathcal{S}} \hat{a}_{t,i} \mathbf{v}_i, \quad \text{where} \quad \hat{a}_{t,i} = \frac{\exp(s_{t,i})}{\sum_{k \in \mathcal{S}} \exp(s_{t,k})}. \tag{11}$$

Let $\epsilon_t = \sum_{j \in \mathcal{E}} a_{t,j}$ denote the *evicted probability mass* (the sum of original attention scores of the evicted tokens). Note that the relationship between the new and original weights is $\hat{a}_{t,i} = \frac{a_{t,i}}{1 - \epsilon_t}$.

**Proposition A.1.** *(Error Bound). Assuming the value vectors are bounded by $\|\mathbf{v}_i\|_2 \leq C$, the approximation error $\|\mathbf{o}_t - \hat{\mathbf{o}}_t\|_2$ is bounded by a term strictly proportional to the evicted mass $\epsilon_t$:*

$$\|\mathbf{o}_t - \hat{\mathbf{o}}_t\|_2 \leq 2C\epsilon_t. \tag{12}$$

*Proof.* We decompose the original output $\mathbf{o}_t$ into the retained part and the evicted part:

$$\mathbf{o}_t = \sum_{i \in \mathcal{S}} a_{t,i} \mathbf{v}_i + \sum_{j \in \mathcal{E}} a_{t,j} \mathbf{v}_j. \tag{13}$$

The approximated output can be rewritten in terms of the original scores:

$$\hat{\mathbf{o}}_t = \sum_{i \in \mathcal{S}} \frac{a_{t,i}}{1 - \epsilon_t} \mathbf{v}_i = \frac{1}{1 - \epsilon_t} \sum_{i \in \mathcal{S}} a_{t,i} \mathbf{v}_i. \tag{14}$$

Now, let $\mathbf{h}_{\mathcal{S}} = \sum_{i \in \mathcal{S}} a_{t,i} \mathbf{v}_i$ and $\mathbf{h}_{\mathcal{E}} = \sum_{j \in \mathcal{E}} a_{t,j} \mathbf{v}_j$. The error vector is:

$$\mathbf{o}_t - \hat{\mathbf{o}}_t = (\mathbf{h}_{\mathcal{S}} + \mathbf{h}_{\mathcal{E}}) - \frac{1}{1 - \epsilon_t} \mathbf{h}_{\mathcal{S}} \tag{15}$$

$$= \mathbf{h}_{\mathcal{E}} + \left(1 - \frac{1}{1 - \epsilon_t}\right) \mathbf{h}_{\mathcal{S}} \tag{16}$$

$$= \mathbf{h}_{\mathcal{E}} - \frac{\epsilon_t}{1 - \epsilon_t} \mathbf{h}_{\mathcal{S}}. \tag{17}$$

Taking the norm and applying the triangle inequality:

$$\|\mathbf{o}_t - \hat{\mathbf{o}}_t\|_2 \leq \|\mathbf{h}_{\mathcal{E}}\|_2 + \frac{\epsilon_t}{1 - \epsilon_t} \|\mathbf{h}_{\mathcal{S}}\|_2. \tag{18}$$

Using the bound $\|\mathbf{v}\| \leq C$, we can infer that:

$$\|\mathbf{h}_{\mathcal{E}}\|_2 = \|\sum_{j \in \mathcal{E}} a_{t,j} \mathbf{v}_j\|_2 \leq \sum_{j \in \mathcal{E}} a_{t,j} C = C\epsilon_t, \tag{19}$$

$$\|\mathbf{h}_{\mathcal{S}}\|_2 = \|\sum_{i \in \mathcal{S}} a_{t,i} \mathbf{v}_i\|_2 \leq \sum_{i \in \mathcal{S}} a_{t,i} C = C(1 - \epsilon_t). \tag{20}$$

Substituting these back yields the upper bound:

$$\|\mathbf{o}_t - \hat{\mathbf{o}}_t\|_2 \leq C\epsilon_t + \frac{\epsilon_t}{1 - \epsilon_t} \cdot C(1 - \epsilon_t) = C\epsilon_t + C\epsilon_t = 2C\epsilon_t. \tag{21}$$

This confirms that the error is bounded by $2C\epsilon_t$. $\qquad \square$

**Implication of the Single-Query Bound.** Proposition A.1 is stated for a single query and shows that the perturbation of the attention output is controlled by the evicted attention mass $\epsilon_t = \sum_{j \in \mathcal{E}} a_{t,j}$. Therefore, at the single-query level, a good eviction strategy should minimize the attention mass removed from the cache.

**From Single Queries to Future Query Blocks.** However, the actual eviction decision in Golden Eviction is not made for one future query at a time. Instead, it is made once at eviction step $t$ and must account for a block of future queries. To bridge this gap, we next introduce a block-level surrogate and show that the future score used in Eq. 5 provides a principled upper bound on the cumulative evicted attention mass over future query blocks.

At eviction step $t$, let $E_t$ denote the evicted set. Since Eq. 5 is defined over future query blocks, let $Q_b$ be the $b$-th future query block and define the pooled attention from block $b$ to cached token $i$ as

$$\bar{a}_{i,b} = \frac{1}{|Q_b|} \sum_{q \in Q_b} a_{q,i}. \tag{22}$$

The corresponding block-level evicted mass is

$$\bar{\epsilon}_{b,t} = \sum_{i \in E_t} \bar{a}_{i,b}. \tag{23}$$

Eq. 5 defines the future score of cached token $i$ at eviction step $t$ as

$$\alpha_{i,t} = \max_{b \geq t} \bar{a}_{i,b}. \tag{24}$$

Thus, for every future block $b \geq t$, we have $\bar{a}_{i,b} \leq \alpha_{i,t}$. Let $H_t$ be the number of future query blocks after eviction step $t$. The cumulative future block-averaged evicted mass is then upper bounded by

$$\sum_{b \geq t} \bar{\epsilon}_{b,t} = \sum_{b \geq t} \sum_{i \in E_t} \bar{a}_{i,b} \tag{25}$$

$$\leq \sum_{b \geq t} \sum_{i \in E_t} \alpha_{i,t} = H_t \sum_{i \in E_t} \alpha_{i,t}. \tag{26}$$

Therefore, selecting the evicted set $E_t$ with the smallest $\sum_{i \in E_t} \alpha_{i,t}$ minimizes an upper bound on the cumulative future block-averaged evicted attention mass. Equivalently, retaining the KV pairs with the largest future scores, as done by Golden Eviction, minimizes this surrogate objective.

**Connection to Generation Quality.** The surrogate above is directly connected to generation quality. Returning from the block level to a particular future query position $q$, Proposition A.1 implies that, under the bounded-value assumption $\|\mathbf{v}_i\|_2 \leq C$, the attention-output perturbation satisfies

$$\|\mathbf{o}_q - \hat{\mathbf{o}}_q\|_2 \leq 2C\epsilon_q, \tag{27}$$

where $\epsilon_q$ is the evicted attention mass for query $q$. If the downstream mapping from attention outputs to logits is locally Lipschitz, then there exists a constant $L_f$ such that

$$\|\mathbf{z}_q - \hat{\mathbf{z}}_q\|_2 \leq L_f \|\mathbf{o}_q - \hat{\mathbf{o}}_q\|_2. \tag{28}$$

If the token loss is also locally Lipschitz with respect to logits, then there exists a constant $L_\ell$ such that

$$|\ell_q - \hat{\ell}_q| \leq L_\ell \|\mathbf{z}_q - \hat{\mathbf{z}}_q\|_2 \tag{29}$$

$$\leq 2C L_f L_\ell \epsilon_q. \tag{30}$$

Therefore, Eq. 5 minimizes the surrogate quantity $\sum_{i \in E_t} \alpha_{i,t}$, which upper-bounds the cumulative future block-level evicted attention mass up to the factor $H_t$. This upper bound propagates to future attention-output perturbation and, under the local stability assumption, to future token-loss drift. This explains why Eq. 5, although defined as a one-step surrogate based on future block-level attention, is well aligned with generation quality preservation.

## B. Model Architectures and Sampling Algorithm

### B.1. Model Design

In KV cache eviction, the design of a scoring model to accurately assess the importance of each key-value pair is crucial. To balance computational efficiency and accuracy, we use an MLP to score the importance of each KV pair. To more fully leverage existing information for prediction, we select two types of information as input:

- *Attention Features*: Inspired by H2O (Zhang et al., 2023) and SnapKV (Li et al., 2024), we use attention scores from both recent windows and cumulative history. For recent windows, we adopt sizes of 8, 16, and 32, along with a window size equal to the update length. For cumulative history, we aggregate attention scores across chunks using two variants: a direct sum and a decayed sum with a per-chunk decay factor of 0.9. For GQA, we concatenate the attention features of all heads within a group as the final attention feature $\mathbf{a}_n \in \mathbb{R}^{6g}$.

- *KV Representations*: We directly concatenate the hidden states of keys $\mathbf{k}_n \in \mathbb{R}^D$ and values $\mathbf{v}_n \in \mathbb{R}^D$ without additional transformation.

Thus, the input feature $\mathbf{x}_n = \text{Concat}([\mathbf{k}_n, \mathbf{v}_n, \mathbf{a}_n]) \in \mathbb{R}^{6g+2D}$. Subsequently, the input feature is sent into the MLP model $\pi_\phi$ to obtain the importance scores:

$$\phi_n = \pi_\theta(\mathbf{x}_n) = \sigma(\mathbf{x}_n \mathbf{W}_1 + \mathbf{b}_1)\mathbf{W}_2 + \mathbf{b}_2, \tag{31}$$

where $\mathbf{W}_1 \in \mathbb{R}^{(6g+2D)\times H}$, $\mathbf{W}_2 \in \mathbb{R}^{H\times 1}$, $\mathbf{b}_1 \in \mathbb{R}^H$, $\mathbf{b}_2 \in \mathbb{R}^1$, $H$ is the immediate size.

### B.2. Sampling Algorithm

After computing the importance score of each KV pair, we apply a Top-$K$ multinomial sampling strategy to determine which pairs to evict. The procedure consists of two steps: (i) we first construct a candidate set by selecting the $K'$ pairs with the lowest predicted importance scores; (ii) we then normalize their negative scores with a softmax function and perform multinomial sampling to select $K$ pairs for eviction. This strategy avoids committing deterministically to potentially noisy local rankings while preventing unrestricted multinomial sampling from evicting highly important KV pairs. As shown in Section 4.3, it outperforms both standard top-$K$ and pure multinomial sampling, while introducing controlled stochasticity that encourages exploration and is particularly suitable for reinforcement learning settings.

## C. Analysis of ForesightKV for Capturing Attention Patterns

As discussed in Section 2.1, the complex attention patterns can not be easily captured by the KV cache eviction algorithms. To evaluate how different KV cache eviction algorithms capture attention patterns, we measure the average similarity of each attention head's outputs before and after applying KV cache compression, as shown in the Table 8. We can observe that, compared with other methods, ForesightKV results in smaller changes in the outputs of attention heads. This indicates that our method can better capture attention patterns and more closely approximate the original distribution, thereby achieving better performance.

*Table 8.* Averaged cosine similairty of attention outputs before and after KV cache eviction.

| Budgets | ForesightKV | RKV | SnapKV | H2O |
|---------|-------------|--------|--------|--------|
| 1K | 0.9736 | 0.9628 | 0.9620 | 0.9711 |
| 2K | 0.9889 | 0.9782 | 0.9816 | 0.9845 |

## D. Analysis of Low-Entropy Tokens

We take samples from the reasoning traces and use Qwen3-4B with R-KV to predict the token at each position based on the maximum probability on math tasks, and then compare these predictions with the original ones. We present the token changes of the top loss increase in low-entropy tokens in Table 9. We can observe that these tokens are often related to factual errors, *e.g.,* numbers and symbols. These errors may influence the following reasoning results.

*Table 9.* Error predictions of low-entropy tokens. The red and blue denote the predicted and original tokens in the sentence.

| | |
|---|---|
| Golden | $8 \times 9 \times 5 \times 7 = 2520$. |
| Prediction | $8 \times 9 \times 5 \times 7 = 8520$. |
| Golden | z = f(x)f(y) = (-1)(-1) = 1. |
| Prediction | z = f(x)f(y) + (-1)(-1) = 1. |

# E. Experiments Details

### E.1. Training Setup

**Data Preparation.**   We begin by preparing the training data using the Qwen3-4B model to generate reasoning traces on the STILL dataset (Min et al., 2024). The generation process employs the same settings as those used during the evaluation phase. From the generated outputs, we select only the corrected traces whose lengths exceed 4096 tokens. This curated collection of long, corrected reasoning traces serves as the dataset for the subsequent supervised training and reinforcement learning stages. All the scoring models for each LLM are trained on the same datasets independently.

**Supervised Training.**   Following data preparation, we conduct supervised fine-tuning on the filtered dataset. The training is performed with a batch size of $8$ and a learning rate of $1e^{-2}$ for a total of $1000$ steps, utilizing a Cosine learning rate scheduler to anneal the learning rate. For our proposed method, we set the hyperparameter $m$ for the Pairwise Ranking Loss to $0.01$. The budget $B$ is configured to $1024$, and the eviction length $L$ is set to $256$.

**Reinforcement Learning.**   In the reinforcement learning phase, we configure the training with a batch size and a mini-batch size of $32$. We employ a fixed learning rate of $3e^{-4}$ with a $10$ step of warmup and train the model for $200$ steps. For each sample in a batch, we generate $8$ eviction trajectories, and each batch of data is trained for one epoch without a mini-batch size. To balance efficiency and performance, the budget $B$ and eviction length $L$ are set to either $(1024,256)$ or $(2048, 512)$, conditional on whether the sequence length is shorter or longer than 12K tokens. The KL penalty coefficient $\beta$ is set to $0.01$ for Qwen3-4B, $0.03$ for Qwen3-1.7B, and $0.1$ for DeepSeek-R1-Distill-Qwen-7B. Finally, the margin $\eta$ in the reward function is set to $1.5$ while $\epsilon$ is set to $0.2$.

**Training Cost.**   For DeepSeek-R1-Distill-Qwen-7B under the 32K setting, the approximate wall-clock cost of the two-stage training on H800 GPUs is about $8\text{h} \times 2$ GPUs for the supervised stage and about $6\text{h} \times 8$ GPUs for the reinforcement learning stage.

### E.2. Baselines

We set H2O (Zhang et al., 2023), SnapKV (Li et al., 2024), and R-KV (Cai et al., 2025) as baselines.

- *H2O*. H2O uses cumulative attention scores as the evaluation metric. At each step, the one with the highest cumulative attention score is retained. To maintain consistency, we also keep a budget of $B$ and perform pruning every $L$ tokens.

- *SnapKV*. SnapKV is a KV cache compression method for long inputs, which uses the last segment of queries as an observation window and determines importance based on their attention scores. Cai et al. (2025) transforms this into a compression method targeting long outputs, assessing the importance of the current step every $L$ tokens by utilizing the attention scores of the final 8 tokens.

- *R-KV*. Based on the aforementioned SnapKV, R-KV also takes into account the redundancy among tokens. It weighs the metrics of window attention score and token-to-token similarity to serve as the importance judgment for a KV pair. The weights for each are 0.1 and 0.9, respectively.

### E.3. Reward Changes Across Training

We present the changes of loss $\mathcal{L}_{\text{ours}}$ with the different training steps. As shown in Figure 4, we can observe that during the initial training phase, the model exhibited unstable loss behavior, fluctuating relative to the original model. After a certain

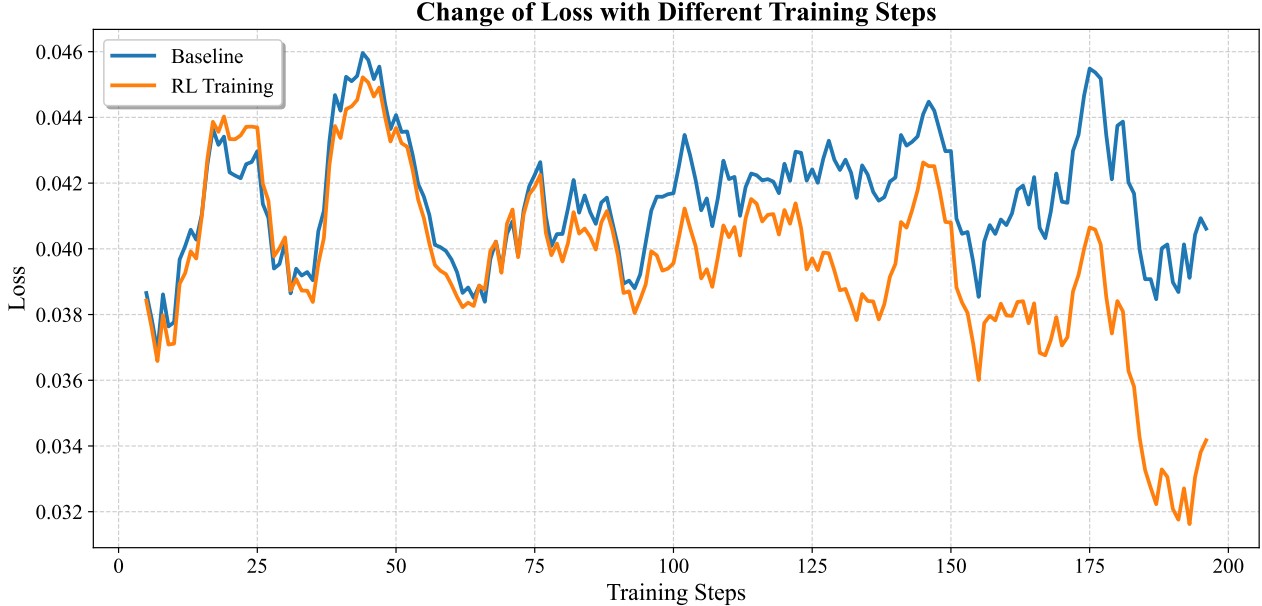

*Figure 4.* Change of losses with different training steps.

number of training steps, the loss consistently decreased, ultimately yielding better results. This indicates that the policy models can achieve effective exploration to improve the reasoning qualities.

### E.4. Reward Definition and Entropy Threshold Sensitivity

Table 4 already compares several reward variants, where the adopted reward achieves the best or tied-best performance in the main setting. We further conduct two sensitivity studies to analyze the robustness of the entropy threshold and the reward definition.

First, we vary the low-entropy threshold on Qwen3-4B with a 1K KV budget and evaluate on AIME2024. Specifically, we compare using all tokens, the lowest-entropy 50% tokens, and the lowest-entropy 80% tokens for reward computation. As shown in Table 10, using all tokens still improves the performance, but is slightly worse than focusing on low-entropy tokens only. In contrast, using only the lowest-entropy 50% tokens performs worse because too few tokens contribute to the reward, which weakens the training signal. The lowest-entropy 80% setting achieves the best performance.

*Table 10.* Sensitivity analysis of the low-entropy threshold on Qwen3-4B with a 1K KV budget, evaluated on AIME2024.

| Threshold | All (100%) | Low 50% | Low 80% |
|---|---|---|---|
| AIME24 | 52.5 | 51.0 | **54.5** |

Second, we compare $\mathcal{L}_{\text{low,large}}$ and $\mathcal{L}_{\text{ours}}$ across different model families and evaluation domains. This comparison is intended to test whether the advantage of the adopted reward only holds in the main math-reasoning setting, or whether it also transfers to other reasoning tasks and backbone models. As shown in Table 11, $\mathcal{L}_{\text{ours}}$ consistently outperforms $\mathcal{L}_{\text{low,large}}$ under both settings. On the science-domain setting, where Qwen3-4B is trained on SCP-116K and evaluated on GPQA, $\mathcal{L}_{\text{ours}}$ improves the score from 46.71 to 49.68. On the math-domain setting with DeepSeek-R1-Distill-Qwen-7B evaluated on AIME2024, it also yields a positive gain from 44.0 to 44.6. These results suggest that explicitly incorporating our reward formulation provides a more reliable training signal than only emphasizing low-entropy tokens with large changes, and that the reward design is not specific to one model or one domain.

Overall, these results support both the robustness of the entropy threshold and the effectiveness of the adopted reward design across models and tasks.

*Table 11.* Comparison of reward definitions across models and domains. All experiments use a 1K KV budget. Higher values indicate better task performance.

| Setting | $\mathcal{L}_{\text{low,large}}$ | $\mathcal{L}_{\text{ours}}$ |
|---|---|---|
| Qwen3-4B / Science: trained on SCP-116K, tested on GPQA | 46.71 | **49.68** |
| DeepSeek-R1-Distill-Qwen-7B / Math: tested on AIME2024 | 44.0 | **44.6** |

## E.5. Length Transfer

In the main setting, the training data uses a maximum sequence length of 32K. To evaluate length transfer, we train another model using data truncated to a maximum length of 8192 tokens, and test it under the same AIME2024 setting with a 1K KV budget. As shown in Table 12, training with the shorter 8192-token maximum length achieves a score of 52.1, which is comparable to the 51.7 score obtained with 32K-length training. This result suggests that ForesightKV does not rely on exactly matching the training and evaluation length scales, and that the learned future-aware eviction behavior can generalize across sequence lengths.

*Table 12.* Length-transfer evaluation on AIME2024 with a 1K KV budget.

| Training Max Length | 8K | 32K |
|---|---|---|
| AIME2024 | **52.1** | 51.7 |

## E.6. Experiments with MiniCPM-4.1-8B

To verify the generalizability of our method beyond the Qwen series, we further evaluate it on MiniCPM-4-1-8B (Xiao et al., 2025a) with a context length of 32K, as detailed in Table 13. The results confirm that our approach consistently outperforms baseline methods on this architecture. Notably, due to the significantly longer generation lengths inherent to MiniCPM compared to the Qwen models, we observe that a larger KV cache budget is requisite to maintain optimal performance.

*Table 13.* Performance evaluation of MiniCPM-4.1-8B on AIME 24 benchmark.

| Method | 2K | 4K | 8K |
|---|---|---|---|
| Full | | 63.3 | |
| R-KV | 13.3 | 36.7 | 50.0 |
| SnapKV | 10.0 | 33.3 | 53.3 |
| ForesightKV(*w/o* RL) | 43.3 | 53.3 | 63.3 |

## E.7. Scalability to Larger Models

We further evaluate whether the proposed mechanism scales beyond the 4B–7B model range used in the main experiments. To this end, we conduct additional experiments on Qwen3-14B and supplementary analyses on Qwen3-14B and Qwen3-32B. These results show that ForesightKV is not limited to smaller backbones, and that the same qualitative trends continue to hold at larger model scales.

First, we evaluate Qwen3-14B on AIME2024 with a 2048 KV budget. As shown in Table 14, ForesightKV substantially outperforms R-KV under the same budget. In addition, ForesightKV without RL already achieves strong performance, while RL further brings a consistent improvement. This is consistent with our interpretation in the main text: the major gain comes from the future-aware eviction design, and the RL stage further refines the learned policy.

Second, we compare different eviction methods on Qwen3-14B and Qwen3-32B using the LM loss ratio relative to FullKV. As shown in Table 15, the trend is consistent with what we observe on Qwen3-4B: Golden Eviction remains much closer to FullKV, while H2O, SnapKV, and R-KV incur clearly larger loss increases. This suggests that future-aware supervision continues to better capture long-term KV importance even at larger model scales.

Finally, we analyze entropy buckets on Qwen3-14B and Qwen3-32B. As shown in Table 16, the same pattern as on

*Table 14.* Scalability evaluation on Qwen3-14B with a 2048 KV budget, evaluated on AIME2024.

| Model | Budget | R-KV | ForesightKV(*w/o* RL) | ForesightKV |
|---|---|---|---|---|
| Qwen3-14B | 2048 | 43.6 | 72.9 | **73.8** |

*Table 15.* LM loss ratio relative to FullKV on larger models.

| Model | Golden Eviction / FullKV | H2O / FullKV | SnapKV / FullKV | R-KV / FullKV |
|---|---|---|---|---|
| Qwen3-14B | **1.0447** | 1.1859 | 1.2191 | 1.2721 |
| Qwen3-32B | **1.0315** | 1.1355 | 1.1581 | 1.2052 |

Qwen3-4B still holds: under R-KV, low-entropy tokens suffer substantially larger loss increases than high-entropy tokens. This observation supports our core motivation that low-entropy tokens can be disproportionately important for preserving reasoning quality, and that this phenomenon is not specific to small models.

*Table 16.* Lm loss of tokens of different entropies under R-KV on larger models.

| Model | All Tokens / FullKV | Top 20% Entropy Tokens / FullKV | Bottom 80% Entropy Tokens / FullKV |
|---|---|---|---|
| Qwen3-14B | 1.3039 | 1.2397 | 1.4646 |
| Qwen3-32B | 1.2300 | 1.1944 | 1.3081 |

Overall, these results support the conclusion that ForesightKV's core mechanism generalizes beyond the original model scale. Both the task performance on Qwen3-14B and the loss-ratio analyses on Qwen3-14B/Qwen3-32B indicate that future-aware eviction remains effective as the backbone model becomes larger.

### E.8. Experiments with more Methods

To further evaluate the effectiveness of ForesightKV, we conduct a comprehensive comparison with G-KV (Liao et al., 2025), Duo-Attention (Xiao et al., 2025b), and RPC (Song et al., 2025). Specifically, we compare ForesightKV with RPC on Qwen3-4B, while comparisons with G-KV and Duo-Attention are performed on DeepSeek-R1-Distill-Qwen-7B. The results on AIME2024 are summarized in Table 17. The results for G-KV are directly taken from the original paper, and for Duo-Attention, we report performance under compression rates of 25% and 50%. All the experiment settings are the same for different methods. As shown in the Table 17, ForesightKV consistently and substantially outperforms all baseline methods across the evaluated settings, demonstrating its superior effectiveness under comparable or even more aggressive compression scenarios.

### E.9. Throughput Analysis

**End-to-End Speed.** On DeepSeek-R1-Distill-Qwen-7B, we evaluate the whole cost under the batch size of 8, and each sample is tested 8 times. It takes 8 and 9.5 hours under the budget of 1K and 2K, and 17.5 hours for the full KV cache. Thus, even under the same budget, our method can achieve about 2 times speedup.

**Speed of KV Cache Eviction.** We evaluate our method on Qwen3-4B with a budget of 2048. Specifically, we generate sequences of length 32K with a batch size of 64, which takes a total of 5813 seconds. Among this, our KV cache eviction mechanism accounts for only 157 seconds, representing just 2.7% of the total time, which is effectively negligible. In contrast, the computation of R-KV takes about 8.1% of the total time, which is much larger. Moreover, due to the larger degree of parallelism, reduced attention computation, and lower cache storage overhead, our method achieves significantly higher computational efficiency, which is sufficient to cover the overhead of the scoring model.

*Table 17.* Performance comparison with more methods.

| Model | Method | 1K ($< 25\%$) | 2K ($< 50\%$) | 4K |
|---|---|---|---|---|
| Qwen3-4B | RPC | 10.0 | 36.7 | 59.2 |
| | ForesightKV | 54.5 | 70.2 | 69.2 |
| DeepSeek-R1-Distill-Qwen-7B | DuoAttention | 3.3 | 13.3 | - |
| | G-KV | 33.0-35.0 | 47.0-49.0 | - |
| | ForesightKV | 44.6 | 52.9 | 54.3 |

