# OpenReview forum: "ForesightKV: Optimizing KV Cache Eviction for Reasoning Models by Learning Long-Term Contribution"
_ICML.cc/2026/Conference — ICML 2026 regular_

### Official Review · Reviewer_Z8kJ · 2026-03-10

**Soundness:** 2
**Presentation:** 3
**Significance:** 2
**Originality:** 2
**Overall Recommendation:** 4
**Confidence:** 4

**Summary:**

This paper proposes ForesightKV, a training-based KV cache eviction framework to improve the efficiency of long-context reasoning in large language models. The method learns a lightweight scoring model to predict the long-term importance of KV pairs and dynamically evicts less important entries under a fixed cache budget. The framework consists of a two-stage training pipeline: supervised training using constructed golden labels and reinforcement learning to further improve its effectiveness.

**Compliance With Llm Reviewing Policy:**

Affirmed.

**Final Justification:**

This paper proposes ForesightKV, a training-based KV eviction strategy using golden labels.
My main concern was its generalization to unseen settings, which is important compared to training-free methods (e.g., H2O). The authors address this with additional experiments and clarifications, and show that the gains come from the eviction design rather than training alone, with scalability to larger models.
Based on this, I change my initial negative assessment to a positive one in favor of acceptance.
Including the rebuttal results in the final version would further strengthen the paper.

**Key Questions For Authors:**

See Weaknesses.

**Limitations:**

yes

**Strengths And Weaknesses:**

**Strengths**

- The paper is generally well written and easy to follow.
- The paper addresses an important problem: KV cache compression for long reasoning traces, which is crucial for improving the efficiency of modern reasoning models.

**Weaknesses**

**[W1] Limited analysis of context length and training cost**

Recent reasoning models (e.g., R1-Distill-7B) support context lengths beyond 32K tokens, while the experiments in this paper mainly consider relatively short contexts (up to 4K). It remains unclear whether the proposed approach would maintain its effectiveness under significantly longer contexts.
In addition, the training cost of the proposed method raises concerns. Although the scoring model itself is lightweight, the two-stage training pipeline (supervised learning + reinforcement learning) may become increasingly expensive as the context length grows. The paper does not provide analysis or discussion of this scalability issue.


**[W2] Potential generalization limitations of the training-based eviction policy**

A key limitation of training-based KV eviction methods is their potential lack of generalization to unseen setups. In this work, the scoring model is trained under fixed training configurations such as context length, cache budget, and eviction interval.
Additional evaluation under more diverse evaluation settings would strengthen the claims, such as:
* cross-domain generalization (e.g., training on math → testing on code or other domains),
* unseen context lengths
* different KV cache budgets and eviction intervals.


**[W3] Limited experimental Scope**

The empirical analysis (Section 2) and the main experiments are conducted primarily on a single model family (Qwen). It remains unclear whether the observed attention patterns and the effectiveness of ForesightKV generalize to other reasoning model families, such as LLaMA-based models.

---

> ### Author Rebuttal · Authors · 2026-03-30
>
> Thank you for raising concrete questions about scalability, generalization, and model coverage. The key clarification is that KV budget and context length are different quantities; several concerns in this review appear to conflate the two. We will make this distinction more explicit in the revision.
>
> ### 1. W1: Context length and training cost
>
> Our experiments are not limited to 4K context. The 4K mentioned in the paper refers to the maximum KV budget in some experiments, not the maximum context length. Both training and evaluation are conducted on 32K long-reasoning settings, so the current results directly demonstrate effectiveness in long-generation scenarios.
>
> For training cost, we will report it more clearly. For DeepSeek-R1-Distill-Qwen-7B on H800 under the 32K setting, the wall-clock cost of the two-stage training is about $8\text{h}\times 2$ GPUs and $6\text{h}\times 8$ GPUs. Meanwhile, Appendix E.6 shows that the end-to-end overhead of eviction is only about 2.7% of total inference time. We will present these numbers more prominently in the revision.
>
> ### 2. W2: Generalization to unseen budget / interval / length / domain settings
>
> Generalization is a central question for training-based eviction methods, and the current draft already covers it along multiple dimensions:
>
> - budget transfer: training uses only KV budgets up to $2K$, but evaluation directly generalizes to a larger $4K$ budget in Figure 3; this is outside the training range, yet the method still maintains stable gains, showing that the learned policy is not tied to a single budget.
> - domain transfer: training uses only math reasoning traces, while evaluation covers science and coding; specifically, we report results on both GPQA and LiveCodeBench in Table 6, showing that the scorer continues to outperform baselines on out-of-domain tasks.
> - length transfer: this is a newly added experiment. We train with a maximum length of 8K and test under the 32K setting; on AIME24 with a 1K budget, it achieves 52.1, close to the original 32K-trained version. This shows that the learned eviction policy is not tied to a single training length and transfers effectively to longer-context settings.
>
> In addition, the supplemented LongBench results show that a policy trained on reasoning traces also transfers to non-reasoning long-input tasks. On this benchmark, ForesightKV reaches an overall average of 38.95, substantially outperforming SnapKV (37.50), R-KV (37.11), and H2O (35.74). We will present this generalization evidence more systematically in the revision.
>
> ### 3. W3: Whether the model family is limited to Qwen
>
> The conclusion is not limited to a single model family. In addition to the Qwen3 series, the current draft also includes DeepSeek-R1-Distill-Qwen-7B in the main experiments and MiniCPM-4.1-8B in the appendix E.4. Although these models differ in training distribution, architectural details, and reasoning behavior, ForesightKV still shows stable gains. This indicates that the attention/eviction patterns identified in the paper are not confined to one model family.
>
> We will make this evidence more prominent in the revision.

---

> > ### Author Rebuttal · Reviewer_Z8kJ · 2026-04-02
> >
> > Thank you for the detailed rebuttal. I will raise my score to 3.
> > While most of my concerns regarding the experimental setup have been clarified, I still have the following additional questions:
> >
> > (a) Source of generalization.
> > The proposed method demonstrates consistent improvements across different budgets / lengths / tasks. However, it remains unclear what primarily drives this effectiveness. Specifically, is the gain mainly due to the benefits of advanced RL-based training (e.g., GRPO), or does it stem from the proposed eviction design itself? A clearer analysis disentangling these factors would strengthen the contribution.
> >
> > (b) Scalability to larger models.
> > While the method is shown to be effective on relatively small-scale models (4B–7B), there remains a concern regarding its scalability to larger model sizes. In particular, it is unclear whether the learned eviction policy would generalize similarly in larger models which may exhibit different attention dynamics. Providing further discussion or analysis (e.g., expected behavior or training cost) on this aspect would improve the paper.

---

> > > ### Author Response · Authors · 2026-04-06
> > >
> > > Thank you for the follow-up.
> > >
> > > **(a) Source of generalization.**
> > > Our view is that the generalization mainly comes from the **eviction design itself**, while **RL is helpful but not the main source**. More specifically, ForesightKV learns the **long-term contribution of KV pairs to future generation**, rather than a local heuristic tied to a specific budget, length, or task setting. This is exactly why the learned policy transfers across different settings.
> > >
> > > This interpretation is supported by the current results. In Figure 3, **ForesightKV w.o. RL** already consistently outperforms prior baselines across different **KV budgets**. In addition, our current paper and supplementary experiments show that the learned policy also transfers to **unseen longer lengths** and to **out-of-domain tasks** beyond math, including science and coding. If the gain mainly came from RL itself, removing RL should not still yield such stable improvements across these different settings. Instead, the evidence suggests that the main source of generalization is the design of the complete eviction design and learning algorithms. Table 2 and Table 5 further support this: Table 2 shows that **Golden Eviction** provides substantially better supervision than heuristic rules, and Table 5 shows that removing key design choices causes clear degradation.
> > >
> > > Therefore, we view RL as playing a different role: it is **not the primary source of generalization**, but rather a way to further improve the policy under the **distribution shift introduced by online eviction**. In other words, the cross-setting transfer mainly comes from the learned future-aware KV utility, while RL provides an additional but secondary gain by improving online robustness. We will make this point clearer in the revision.
> > >
> > > **(b) Scalability to larger models.**
> > > We agree that scalability to larger models is important. To address this concern, we further added experiments on **Qwen3-14B** and supplementary analyses on **Qwen3-14B / Qwen3-32B**. The new results show that ForesightKV is not limited to the 4B–7B regime, and that the same qualitative trends continue to hold at larger scales.
> > >
> > > First, on **Qwen3-14B** with a **2048 budget**, ForesightKV still substantially outperforms prior methods. Moreover, **ForesightKV w.o. RL** is already very strong, and RL further brings a consistent improvement, which is again consistent with the interpretation above: the main gain comes from the eviction design, while RL provides an additional refinement. The training costs of the two stages are $15\times 2$ and $14 \times 8$ hours, respectively.
> > >
> > > | Model     | Budget | R-KV | ForesightKV w.o. RL | ForesightKV |
> > > | --------- | -----: | ---: | ------------------: | ----------: |
> > > | Qwen3-14B |   2048 | 43.6 |                72.9 |        73.8 |
> > >
> > > Second, we compared different eviction methods on **Qwen3-14B and Qwen3-32B** using the LM loss ratio relative to FullKV. The trend is consistent with what we observe on Qwen3-4B: **Golden Eviction** remains much closer to FullKV, while H2O, SnapKV, and R-KV are clearly worse. This suggests that our future-aware supervision continues to better capture long-term KV importance even at larger model scales.
> > >
> > > | Model     | Golden Eviction / FullKV | H2O / FullKV | SnapKV / FullKV | R-KV / FullKV |
> > > | --------- | -----------------------: | -----------: | --------------: | ------------: |
> > > | Qwen3-14B |                   1.0447 |       1.1859 |          1.2191 |        1.2721 |
> > > | Qwen3-32B |                   1.0315 |       1.1355 |          1.1581 |        1.2052 |
> > >
> > > Finally, we also analyzed entropy buckets on **Qwen3-14B and Qwen3-32B**, and found the same pattern as on Qwen3-4B: under R-KV, **low-entropy tokens suffer substantially larger loss increases than high-entropy tokens**. This shows that our core motivation is not specific to small models, but continues to hold at larger scales as well.
> > >
> > > | Model     | All Tokens / FullKV | Top 20% Entropy Tokens / FullKV | Bottom 80% Entropy Tokens / FullKV |
> > > | --------- | ------------------: | ------------------------------: | ---------------------------------: |
> > > | Qwen3-14B |              1.3039 |                          1.2397 |                             1.4646 |
> > > | Qwen3-32B |              1.2300 |                          1.1944 |                             1.3081 |
> > >
> > > Overall, these new results support the conclusion that **ForesightKV’s core mechanism generalizes beyond the original model scale**. We will include these results in the revision and clarify that the same generalization mechanism appears to hold at larger scales as well.

---

### Official Review · Reviewer_1PQM · 2026-03-11

**Soundness:** 3
**Presentation:** 3
**Significance:** 3
**Originality:** 2
**Overall Recommendation:** 4
**Confidence:** 4

**Summary:**

This paper proposes ForesightKV for KV-cache eviction.

The motivation aligns with the general direction that identify valuable pairs to retain based on their future usefulness. Another analysis focuses on low-entropy tokens, which often encode deterministic information and can induce a loss increase if evicted.

Although the methods are simple and lack theoretical depth, the authors provide very logically consistent validation experiments and method design. For methods, first they construct a training pipeline to update a simple MLP for scoring. Second, they adopt the GRPO loss to capture long-term gains.

**Compliance With Llm Reviewing Policy:**

Affirmed.

**Final Justification:**

As my original opinion was borderline and I gave the paper a 4, the partially complete rebuttal leads me to maintain a positive overall assessment. However, the **large** gain from Top-k multinomial ultimately weakens the paper’s overall soundness. Though the authors claim it is not technically omittable, it is **not** well discussed in the original Methods section.

**Key Questions For Authors:**

See W1 and W2. More baselines are encouraged.

**Strengths And Weaknesses:**

**Strengths**

- S1. Each single step, though the methods I think are easy to design, is reasonable and clearly validated. The paper reconducts the semantic dependency analysis, and also finds that low-entropy tokens are dangerous to remove during cache eviction. These experiments solidly support their framework.
- S2. The presentation is clear, especially the motivation of each method module.
- S3. The implementation is strong so that the overall efficiency gain is stable, which is a general problem for eviction.

**Weaknesses**

- W1. According to Table 5, the gain from Top-k multinomial is extremely high. This induces a very severe concern that the performance comes from this trick. Note that Top-k multinomial is just briefly introduced in the method section, while why this happens (e.g., is Top-k multinomial more suitable for your method design compared with Top-k or multinomial) is not clearly discussed.
- W2. No training time cost. How much time and computation are needed for training and RL?

**Suggestions**

- SG1. As MLP is used as a scorer, it would be interesting to check how important kn, vn, an (in Eq. 1) are when determining the importance of a KV pair.

- SG2. The author highlights that KV cache expands "linearly"; however, in their main experiment, the scale they reduced to is still somehow linear instead of log or something smaller. As a general setting, the experiments are fine, but emphasis on linear expansion could be revised.

- SG3. ϕ in Eq. 6 Page 5 should be mathematically rederived, which is mentioned too early (Eq. 1, Page 3).

---

> ### Author Rebuttal · Authors · 2026-03-30
>
> Thank you for recognizing the design motivation, clarity of presentation, and implementation stability.
>
> ### 1. W1: Whether top-k multinomial dominates the performance gain
>
> Top-k multinomial is not a sampling trick detached from the main method, but a more suitable action parameterization for the learned scorer in the sequential eviction setting, analogous to sampling in LLM generation.
>
> - Pure top-k is similar to greedy decision-making: low variance, but sensitive to local ranking errors.
> - Pure multinomial is similar to direct sampling: stronger exploration, but too noisy, especially since eviction is sequential and irreversible.
> - Top-k multinomial is similar to top-k + top-p sampling: it first restricts actions to a high-confidence candidate set, and then performs controlled sampling within that set, thus achieving a better balance between stability and exploration.
>
> This is beneficial not only for inference-time robustness, but also for RL training. More importantly, the performance gain does not come from this design alone. Table 5 already shows that both pure top-k and pure multinomial are clearly weaker than top-k multinomial, but pure top-k is already significantly stronger than the baselines. At the same time, using only attention features also causes a substantial drop. Therefore, the gains come from the combined effect of the learned scorer, KV-aware representation, RL objective, and this action parameterization, rather than from any single trick.
>
> ### 2. W2: Missing training cost report
>
> We will explicitly add training and deployment costs in the revision.
>
> For DeepSeek-R1-Distill-Qwen-7B on H800 under the 32K setting, the wall-clock cost of the two-stage training is approximately:
>
> - supervised stage: about $8\text{h} \times 2$ GPUs;
> - RL stage: about $6\text{h} \times 8$ GPUs.
>
> In addition, Appendix E.6 already reports the end-to-end overhead: the extra cost of eviction accounts for about 2.7% of total inference time, lower than R-KV's 8.1%. We will move these numbers forward in the revision and add a dedicated cost breakdown table.
>
> ### 3. SG1: Importance of key/value/attention features
>
> We supplemented the scorer input feature ablation:
>
> | Input | AIME24 Score |
> | --- | ---: |
> | Attention | 29.0 |
> | Attention + value | 47.9 |
> | Attention + key | 48.3 |
> | Attention + key + value | 50.6 |
>
> This shows that attention-only is clearly insufficient; adding either key or value brings a substantial improvement, with key performing slightly better, and the combination of all three performing best. This supports the design of the KV-aware scorer.
>
> ### 4. SG2: On the phrasing of "linear KV growth"
>
> For a standard Transformer, the KV cache size itself grows linearly with context length, whereas our method keeps the actual cache within a fixed budget $B$ under the fixed-budget eviction setting, thereby substantially mitigating the practical overhead caused by linear growth. In that sense, the effective cache cost of our method is constant. We will revise the wording accordingly to make this point clearer.
>
> ### 5. SG3: Symbol $\phi$ / notation and derivation order around Eq. (6)
>
> In the original draft, $\phi$ first appears in Eq. (1) as the scorer output, while its relationship to future-importance ordering is not explicitly stated until Eq. (6), which hurts readability.
>
> In the revision, we will reorder this part of the derivation: first explain that Golden Eviction depends only on relative ordering, and then state that the scorer's training objective is to preserve the ordering induced by the future score $\alpha$, namely,
> $$
> \alpha_{i,t}>\alpha_{j,t}\Rightarrow \phi_{i,t}>\phi_{j,t},
> $$
> which naturally leads to the pairwise margin ranking loss.

---

> > ### Author Rebuttal · Reviewer_1PQM · 2026-04-02
> >
> > W1. So please add more discussion about the advantage of Top-k multinomial in the method section instead of the experiment part.
> >
> > SG2. Mentioning the constant cost when introducing related works is preferred.

---

> > > ### Author Response · Authors · 2026-04-02
> > >
> > > Thank you for the helpful follow-up. We agree with both suggestions and will revise the manuscript accordingly.
> > >
> > > **Regarding W1 (discussion of Top-k multinomial).**
> > > We agree that the motivation for **Top-k multinomial** should be explained directly in the **method section**, rather than relying mainly on the ablation results. In the revised manuscript, we will expand the discussion around **Eq. (2)** and add the following clarification:
> > >
> > > > *We use Top-k multinomial rather than pure Top-k or pure multinomial because KV eviction is sequential and irreversible. Pure Top-k is overly greedy and sensitive to local score errors, while pure multinomial introduces excessive variance. Top-k multinomial first restricts the action space to a high-confidence candidate set and then performs controlled sampling within that set, achieving a better balance between stability and exploration. In addition, this parameterization is shared by both RL training and final evaluation, which reduces the mismatch between training-time policy exploration and inference-time eviction behavior.*
> > >
> > > We will also revise the discussion of **Table 5** so that it serves as empirical support for this design choice, rather than the main place where the rationale is introduced.
> > >
> > > **Regarding SG2 (introducing the constant-cost point earlier).**
> > > We also agree that the “constant effective cache cost” point should be stated earlier. In the revised manuscript, we will revise the related-work discussion starting from the paragraph describing existing eviction methods (i.e., *“During the generation process, these methods typically employ elaborately designed rules ...”*) in the Introduction Section and clarify that, although the raw KV cache of a standard Transformer grows linearly with context length, under the fixed-budget eviction setting the retained cache is always bounded by the preset budget throughout decoding. Therefore, the effective retained-cache cost of our method is constant with respect to generation length.
> > >
> > > We thank the reviewer again for these constructive suggestions. We believe these changes will improve the clarity of the paper.

---

### Official Review · Reviewer_24Cg · 2026-03-12

**Soundness:** 3
**Presentation:** 3
**Significance:** 3
**Originality:** 3
**Overall Recommendation:** 4
**Confidence:** 3

**Summary:**

This paper proposes a method called ForesightKV. It is a training-based KV cache eviction framework that learns to predict which KV pairs to evict during long-text generation, thereby speeding up long generation. Compared to other existing evication methods, which often failed due to the reason that kv importance can be global, position-based, or semantic and time-varying. Their experiments show they can outperform other eviction methods.

**Compliance With Llm Reviewing Policy:**

Affirmed.

**Final Justification:**

I thank the authors for the rebuttal. I think my main concern was about the benchmark coverage, and I think the added LongBench results address this well. They strengthen the paper by showing the method also works in long-context settings beyond reasoning tasks. Overall, my assessment remains positive, and I keep my original score.

**Key Questions For Authors:**

The critical question is that only math and science-related benchmarks are not enough.

**Limitations:**

I think the papers need to include more benchmarks.

**Strengths And Weaknesses:**

Strengths:
- Good explanation. The paper analyzes three different kv importance patterns and discusses the necessity of the methods.
- Clear writing. The paper clearly describes their method, experiments and etc.

Weakness:
- One concern is that the paper only focuses on reasoning tasks. Thus, the trained MLP scorer might be a better fit for reasoning tasks, but not for tasks like factual memorization, etc.
- The paper might need some evaluations not only on reasoning tasks, but also some long context tasks and reports the accuracy or some retrival related score.

---

> ### Author Rebuttal · Authors · 2026-03-30
>
> Thank you for emphasizing the importance of broader benchmarks, especially non-reasoning and long-context settings.
>
> First, the current draft already includes cross-domain generalization results: training uses only math reasoning traces, while testing covers science and coding, showing that the learned scorer is not restricted to a single task domain.
>
> Second, we added non-reasoning long-input evaluation on LongBench. Under this setting, ForesightKV achieves an overall average of 38.95, substantially outperforming SnapKV (37.50), R-KV (37.11), and H2O (35.74), while being only 0.46 below the full model (39.41). This shows that even on a non-reasoning long-input benchmark, ForesightKV achieves a better quality-compression tradeoff under a fixed KV budget, rather than being effective only for math reasoning.
>
> | Task Category | Full Model | R-KV | SnapKV | H2O | ForesightKV |
> | --- | ---: | ---: | ---: | ---: | ---: |
> | Single-document QA | 41.88 | 37.39 | 37.29 | 35.26 | **41.07** |
> | Multi-document QA | 43.97 | 42.45 | 42.56 | 40.51 | **43.46** |
> | Summarization | 27.48 | 24.08 | 24.97 | 21.13 | **27.39** |
> | Few-shot Learning | 62.73 | 59.39 | **60.71** | 59.27 | 60.48 |
> | Synthetic Tasks | 48.88 | **50.02** | 50.02 | 48.50 | 49.55 |
> | Code Completion | 2.29 | 1.91 | 1.64 | 3.17 | **3.41** |
> | Overall Average | 39.41 | 37.11 | 37.50 | 35.74 | **38.95** |
>
> Therefore, although our method is trained on long-output reasoning traces, the learned eviction policy does not overfit to a single task format, but transfers to more general long-context tasks. We will place these results in a more prominent position in the revision and strengthen the related discussion.

---

> > ### Author Rebuttal · Reviewer_24Cg · 2026-04-02
> >
> > The author fully resolved my concern. I will keep my positive score

---

### Official Review · Reviewer_44Fw · 2026-03-18

**Soundness:** 3
**Presentation:** 3
**Significance:** 3
**Originality:** 3
**Overall Recommendation:** 4
**Confidence:** 3

**Summary:**

This paper studies KV cache eviction for long-form reasoning models, where cache growth creates substantial memory and latency overhead during decoding. The authors propose ForesightKV, a training-based eviction framework that learns a lightweight scoring model to predict the long-term contribution of KV pairs and evict less useful ones during generation. The method has two stages: first, a supervised stage based on a proposed Golden Eviction algorithm that constructs future-aware eviction labels from attention traces; second, a reinforcement learning stage that formulates eviction as an MDP and optimizes the scorer using GRPO with rewards designed around loss increases on low-entropy tokens. Experiments on AIME2024/AIME2025 with Qwen3-1.7B, Qwen3-4B, and DeepSeek-Qwen-7B show that ForesightKV outperforms prior KV eviction baselines under the same cache budget and often achieves similar performance with roughly half the cache.

**Compliance With Llm Reviewing Policy:**

Affirmed.

**Final Justification:**

I will keep my current evaluation since the observed weaknesses would be hard to resolved and I think the current positive evaluation would be enough.

**Key Questions For Authors:**

1. How sensitive is ForesightKV to the reward definition and entropy threshold across model families and non-math domains? I think more analysis would be super helpful.
2. See the weaknesses above.

**Limitations:**

Yes

**Strengths And Weaknesses:**

Strengths:
1. The writing of the paper is clear and the motivation is easy to follows.
2. The paper tackles a relevant systems problem for reasoning LLMs: decoding-time KV cache growth is a real bottleneck for long CoT generation, and improving the quality-efficiency tradeoff here is important.
3. The authors provide proof to support the golden eviction algorithm, showing that this is a meaningful step beyond rule-based methods. And the empirical observations in Section 2 are useful motivation: the paper argues that attention patterns can be global, position-dependent, or semantic-dependent, and that low-entropy tokens are especially fragile under eviction.

Weaknesses:
1. The main technical justification for the supervised target seems somewhat indirect. The “Golden Eviction” objective is motivated via future attention scores and a bound in the appendix, but in the main paper the connection between minimizing future attention and preserving downstream generation quality remains only partially established. The empirical evidence is good, but the theoretical justification in the main text can be further explored.
2. The evaluation is somewhat narrow in task diversity for the main claim. The most results mostly focused on AIME datasets. I think the evaluation results on more types of tasks can strengthen the conclusions of the paper.

---

> ### Author Rebuttal · Authors · 2026-03-30
>
> Thank you for your insightful suggestions!
>
> ### 1. On the theoretical connection between future attention and generation quality
> We agree that Eq. (5) is not merely heuristic. It minimizes a surrogate objective that upper-bounds the future attention mass lost due to eviction at the **block level**, which in turn controls future attention-output perturbation and, under a standard local stability assumption, the drift in future token losses.
>
> At eviction step $t$, let $E_t$ be the evicted set. Since Eq. (5) is defined on future query blocks, let $Q_b$ denote the $b$-th future query block, and define the pooled attention from block $b$ to cached token $i$ as
> $\bar a_{i,b} = \frac{1}{|Q_b|}\sum_{q\in Q_b} a_{q,i}$.
>
> The corresponding block-level evicted mass is
> $\bar\epsilon_{b,t} = \sum_{i\in E_t}\bar a_{i,b}$.
>
> Eq. (5) defines the future score as
> $\alpha_{i,t} = \max_{b\ge t}\bar a_{i,b}$.
>
> Therefore, for every future block $b\ge t$, we have $\bar a_{i,b}\le \alpha_{i,t}$, and thus
> $\sum_{b\ge t}\bar\epsilon_{b,t}
> = \sum_{b\ge t}\sum_{i\in E_t}\bar a_{i,b}
> \le H_t \sum_{i\in E_t}\alpha_{i,t}$,
> where $H_t$ is the number of future query blocks. Hence, minimizing $\sum_{i\in E_t}\alpha_{i,t}$ minimizes an upper bound on the cumulative future **block-averaged** evicted attention mass. This makes Eq. (5) a principled one-step surrogate objective rather than a purely heuristic rule.
>
> This surrogate is directly connected to generation quality. From Appendix A, under the bounded-value assumption $\|v_i\|_2\le C$, the attention-output perturbation at a future query position $q$ satisfies
> $\|o_q-\hat o_q\|_2 \le 2C\,\epsilon_q$,
> where $\epsilon_q$ is the evicted attention mass for that query. If the downstream mapping from attention outputs to logits is locally Lipschitz, then there exists $L_f$ such that
>
> $$\|z_q-\hat z_q\|_2 \le L_f \|o_q-\hat o_q\|_2$$
>
> If the token loss is also locally Lipschitz with respect to logits, then there exists $L_\ell$ such that
>
> $$|\ell_q-\hat\ell_q| \le L_\ell \|z_q-\hat z_q\|_2$$
>
> $$\le 2C L_f L_\ell \epsilon_q$$
>
> Since our surrogate operates at the block level, we average this bound over all queries $q \in Q_b$:
>
> $$\frac{1}{|Q_b|} \sum_{q \in Q_b} |\ell_q-\hat\ell_q| \le 2C L_f L_\ell \left( \frac{1}{|Q_b|} \sum_{q \in Q_b} \epsilon_q \right) = 2C L_f L_\ell\, \bar{\epsilon}_{b,t}$$
>
> Therefore, Eq. (5) minimizes a surrogate that upper-bounds cumulative future block-level evicted attention mass. Since this mass controls attention-output perturbation and, under local stability assumptions, the resulting logit and loss drift, Eq. (5) is a principled objective for preserving future generation quality rather than a purely heuristic rule.
>
> ### 2. On task diversity, and why the evaluation should not focus only on AIME
> The method is not limited to math tasks. The current draft already includes cross-domain results on GPQA (science) and LiveCodeBench (coding): although the scorer is trained only on math reasoning traces, it still consistently outperforms baselines under the same KV budget.
>
> We further add non-reasoning long-input evaluation on LongBench. ForesightKV achieves an overall average of 38.95, outperforming SnapKV (37.50), R-KV (37.11), and H2O (35.74), while trailing the full model by only 0.46 points (39.41). This supports generalization beyond math reasoning.
>
> | Full  | R-KV | SnapKV | H2O | ForesightKV |
> | ---: | ---: | ---: | ---: | ---: |
>  | 39.41 | 37.11 | 37.50 | 35.74 | **38.95** |
>
> We will move these cross-domain and LongBench results earlier in the revision.
>
> ### 3. On the reward definition and sensitivity to the entropy threshold
>
> Table 4 already compares several reward variants, and the adopted reward is the best or tied for best in the main setting.
>
> We further add two sensitivity results. First, for the low-entropy threshold, we compare all tokens, the lowest-entropy 50%, and the lowest-entropy 80% on Qwen3-4B / AIME24 / 1K budget. The lowest-entropy 80% performs best. Using all tokens still improves performance, but is slightly worse than focusing on low-entropy tokens only. Using only the lowest-entropy 50% performs worse because too few tokens contribute to the reward, which weakens the training signal.
>
> | Threshold | All  (100%) | Low 50% | Low 80% |
> | --- | ---: | ---: | ---: |
> | AIME24 | 52.5 | 51.0 | 54.5 |
>
> Second, we compare $\mathcal L_{low}$ and $\mathcal L_{ours}$ across models and domains. $\mathcal L_{ours}$ outperforms $\mathcal L_{low}$ in both settings. This shows that the reward design is not specific to one model or one domain.
>
> | Setting | $ L_{low}$ | $ L_{ours}$ |
> | --- | ---: | ---: |
> | Qwen3-4B / Science: trained on SCP-116K, tested on GPQA, 1K budget | 46.71 | 49.68 |
> | DeepSeek-R1-Distill-Qwen-7B / Math: tested on AIME24, 1K budget | 44.0 | 44.6 |
>
> Overall, these results support both the robustness of the entropy threshold and the effectiveness of the adopted reward design across models and tasks.

---

> > ### Author Rebuttal · Reviewer_44Fw · 2026-04-06
> >
> > The rebuttal resolves my questions mostly and the provided explanations could strengthen the conclusions in the paper. I think this paper is a solid one in overall. I will keep my current evaluations with a positive score.

---

> > > ### Author Response · Authors · 2026-04-07
> > >
> > > Thank you very much for reading our rebuttal carefully and for confirming that the main concerns have been addressed. We are glad that our clarifications helped strengthen the paper.
> > >
> > > If you feel the rebuttal has adequately resolved your previous concerns, we would be very grateful if you could kindly increase your score. Thank you again for your time and thoughtful feedback.

---

### Decision · Program_Chairs · 2026-04-30

**Decision:**

Accept (regular)

**Comment:**

The committee unanimously agrees that this paper tackles a highly relevant and critical systems bottleneck for modern reasoning LLMs. The dual-stage methodology is principled and represents a meaningful advancement over purely rule-based or heuristic eviction strategies. During the discussion phase, the authors provided an exceptionally thorough and compelling rebuttal that systematically addressed the reviewers' initial reservations.